# Performance of ChatGPT on Chinese Master's Degree Entrance Examination in Clinical Medicine

**Ke-Cheng Li**[1‡], **Zhi-Jun Bu**[2‡], **Md. Shahjalal**[3], **Bai-Xiang He**[4], **Zi-Fan Zhuang**[5], **Chen Li**[6], **Jian-Ping Liu**[2], **Bin Wang**[1]*, **Zhao-Lan Liu**[2]*

**1** Department of Andrology, Dongzhimen Hospital, Beijing University of Chinese Medicine, Beijing, China,
**2** Centre for Evidence-Based Chinese Medicine, Beijing University of Chinese Medicine, Beijing, China,
**3** Department of Public Health, North South University, Dhaka, Bangladesh, **4** Department of Gastroenterology, Dongzhimen Hospital, Beijing University of Chinese Medicine, Beijing, China,
**5** Department of Endocrinology, Guang'anmen Hospital, China Academy of Chinese Medical Sciences, Beijing, China, **6** Institute of Artificial Intelligence and Robotics, Xi'an Jiaotong University, Xi'an, Shannxi, China

‡ K-CL and Z-JB contributed equally and share first authorship
* lzl1019@163.com (Z-LL); dayiwangbin@sina.com (BW)

**Data Availability Statement:** All relevant data are within the manuscript and its Supporting information files.

## Abstract

### Background

ChatGPT is a large language model designed to generate responses based on a contextual understanding of user queries and requests. This study utilised the entrance examination for the Master of Clinical Medicine in Traditional Chinese Medicine to assesses the reliability and practicality of ChatGPT within the domain of medical education.

### Methods

We selected 330 single and multiple-choice questions from the 2021 and 2022 Chinese Master of Clinical Medicine comprehensive examinations, which did not include any images or tables. To ensure the test's accuracy and authenticity, we preserved the original format of the query and alternative test texts, without any modifications or explanations.

### Results

Both ChatGPT3.5 and GPT-4 attained average scores surpassing the admission threshold. Noteworthy is that ChatGPT achieved the highest score in the Medical Humanities section, boasting a correct rate of 93.75%. However, it is worth noting that ChatGPT3.5 exhibited the lowest accuracy percentage of 37.5% in the Pathology division, while GPT-4 also displayed a relatively lower correctness percentage of 60.23% in the Biochemistry section. An analysis of sub-questions revealed that ChatGPT demonstrates superior performance in handling single-choice questions but performs poorly in multiple-choice questions.

### Conclusion

ChatGPT exhibits a degree of medical knowledge and the capacity to aid in diagnosing and treating diseases. Nevertheless, enhancements are warranted to address its accuracy and

**Funding:** This study is supported by a grant from the National Natural Science Foundation of China (Grant No. 82374298) and the Reserve Discipline Leader Funding of Beijing University of Chinese Medicine (Grant No. 90010960920033). There was no additional external funding received for this study.

**Competing interests:** The authors have declared that no competing interests exist.

reliability limitations. Imperatively, rigorous evaluation and oversight must accompany its utilization, accompanied by proactive measures to surmount prevailing constraints.

## 1. Introduction

A large language model (LLM) is a computer program that employs artificial intelligence and natural language processing technology to comprehend and generate natural language text from extensive data, utilizing deep learning techniques [1]. Developed by OpenAI, ChatGPT is a substantial language model that interacting with users through dynamic dialogues, effectively responding to inquiries and requests. Notably, ChatGPT has garnered significant recognition for its exceptional performance, particularly within the medical domain [2, 3]. Presently, ChatGPT offers users two versions: ChatGPT3.5, available for free, and GPT-4, which requires payment. In comparison to ChatGPT3.5, GPT-4 elevates its model size from 175 billion to an impressive 170 trillion, incorporating a rule-based reward modeling (RBRM) methodology. Furthermore, it refines this methodology through human feedback reinforcement learning of the generated text, employing Reinforcement Learning Fine-tuning with Human Feedback (RLHF). These advancements contribute significantly to the reliability and security of GPT-4 [4, 5].

Impressively, ChatGPT has demonstrated an accuracy rate of 60%, approaching the pass threshold on the United States Medical Licensing Examination (USMLE), without the prerequisite of prior input of relevant background knowledge [6]. Additionally, its performance on the NBME-Free-Step1 dataset from the American Board of Medical Examiners surpasses the 60% pass threshold, indicative of skills comparable to a third-year medical student [7]. This accuracy underscores ChatGPT's solid grasp of medical knowledge and remarkable proficiency in logical reasoning and disease diagnosis. ChatGPT also exhibits variable performance across different medical specializations. For instance, both ChatGPT3.5 and GPT-4 scored below the passing threshold and the average score of nephrology candidates in the American Society of Nephrology (ASN) Nephrology Self-assessement Program and Renal Self-assessement Program tests [8]. Furthermore, ChatGPT displayed an incorrect rate of 66% on PACES, the French medical school entrance exam [9]. While large-scale language models achieve profound semantic comprehension through extensive data and context, their efficacy in meeting user needs in managing intricate or detailed information and processing non-English language input. Moreover, their proficiency in handling medical information in non-English texts is currently insufficient and necessitates further enhancement and development.

In China, prospective Master's degree students are mandated to undertake the Nationwide Master's Program Unified Admissions Examination (NMPUA), a government-organized assessement facilitating entry into their desired Master's programs. The Comprehensive Examination of Clinical Medicine is obligatory for those pursuing a professional Master's degree in clinical medicine, aiming to comprehensively evaluate the clinical performance of undergraduate graduates in clinical medicine within clinical scenarios. NMPUA represents a critically important examination for aspiring master's degree students, aimed at assessesing the clinical reasoning, knowledge, diagnostic capabilities, and decision-making proficiency of medical undergraduates in a clinical context. The examination encompasses five question types: A1 (knowledge-based multiple choice), A2 (case-based multiple choice), A3 (case-group-based multiple choice), B (matching), and X (multiple choice), totaling 165 multiple-choice questions with a maximum score of 300. The exam spans six sections, covering

physiology, biochemistry, pathology, internal and external medicine in clinical medicine, and medical humanities. Each single-choice question features one correct answer and three incorrect interfering options, while each multiple-choice question includes at least two correct answers. All questions and options were presented in text format, eliminating the need to analyze pictures or tables visually. Following rigorous review and screening by two independent researchers, all 330 questions met the study's criteria and were included in the test battery.

The main objective of this study was to assesses the accuracy, robustness, and limitations of ChatGPT3.5 and GPT-4 in the context of the Chinese Master's Comprehensive Examination in Clinical Medicine. This evaluation aimed to ascertain the effectiveness and reliability of ChatGPT within the Chinese medical domain and to provide guidance and references to assist Chinese medical students in their examination preparation. On the one hand, ChatGPT proves beneficial in aiding candidates to identify areas of fundamental knowledge that require improvement, thereby enhancing their performance in the final examination. On the other hand, educators can optimize the utility of ChatGPT by providing candidates access to more specialized test questions accompanied by custom-generated feedback. This approach facilitates greater automation and efficiency in the marking and feedback processes.

## 2. Methods and materials

### 2.1. Data collection

The test battery employed in this study comprised the Chinese Master's Comprehensive Clinical Medicine Examinations (code 306) for the years 2021 and 2022, encompassing a total of 330 questions. These questions were distributed across various types, including 127 A1-type questions, 23 A2-type questions, 80 A3-type questions, 40 B-type questions, and 60 X-type questions. Notably, the Physiology, Biochemistry, and Pathology sections exclusively featured A1, A2, B, and X-type queries, whereas Medical Humanities solely consisted of A1-type questions. Internal Medicine and Surgery encompassed all question types.

The entirety of the paper was composed in Chinese, with occasional inclusion of English acronyms for specific medicines or proper nouns, which were preserved in their original untranslated form. Marks were allocated for questions 1–40 and 116–135 at a rate of 1.5 marks each, while questions 41–115 and 136–165 carried a weight of 2 marks each.

### 2.2. Study design

In this study, we replicated and sent 330 multiple-choice questions from the 2021 and 2022 Chinese Master of Clinical Medicine comprehensive examinations to both ChatGPT3.5 and GPT-4, in the order they appeared in the examination papers. The request was for them to simulate the role of a doctor and provide answers accordingly.

Each question was restricted to a single answer, irrespective of the accuracy of the response. We intentionally refrained from prompting ChatGPT to furnish an analysis of the options or an explanation for its choices. To minimize the influence of extraneous factors on results, participants were instructed to answer multiple-choice questions as medical professionals without offering any justifications. Responses were meticulously recorded using Excel and cross-verified against the correct answers to ensure precise evaluation of their performance in the Masters Comprehensive Clinical Medicine Examination. Through the computation of the percentage of accurate responses and the derivation of scores, our objective elucidate the potential advantages and challenges associated with the utilization of ChatGPT in the application of medical knowledge. The methodology employed in our study seeks to provide a comprehensive understanding of the practicality of ChatGPT's current applications in the medical

domain and shed light on its prospective role in medical education, as well as in the diagnosis and treatment of ailments.

In examining of ChatGPT's performance features, our focus was directed towards understanding the impact of modifying temperature values on the reliability of generated responses. Temperature values play a crucial role in large language models as they directly influence the randomness of the generated content. Generally ranging from 0 to 1, lower temperatures (below 0.3) tend to produce more dependable and consistent outcomes, while higher temperatures (above 0.7) result in more varied and imaginative outputs [10, 11]. Notably, the default temperature setting in ChatGPT is typically 0.7. To gain insights into how temperature adjustments may impact answer correctness, we systematically tested the performance of both ChatGPT3.5 and GPT-4 concerning correctness at four temperature values (0, 0.3, 0.7, and 1). The experiment aimed to unveil how adjusting the temperature influences the reliability and diversity of generated answers using ChatGPT in the medical field. The findings from this study provide valuable insights for model optimization, shedding light on the role of temperature parameters in ChatGPT. Furthermore, it enhances our understanding of ChatGPT's performance tuning in practical applications within the medical field.

We meticulously documented the accurate success rates of ChatGPT across various subjects and question types. The overarching objective was to deepen our understanding of ChatGPT's proficiency in diverse knowledge domains and its ability to address different question types. Special attention was dedicated to evaluating whether ChatGPT adhered to the rules governing the answering of questions. Additionally, questions that received either entirely incorrect or correct responses underwent detailed analysis to comprehensively assesses ChatGPT's answering abilities and distinctive features.

## 2.3. Statistical analysis

The data for this study were collected and analyzed using Microsoft Excel Mac 16.66.1 (Microsoft Corp., USA), and accuracy and scoring rate were presented in percentages. The Python programming language was employed for charting, visualization, and in-depth analysis to enhance the intuition and presentation of the findings.

## 3. Results

At temperatures of 0, 0.3, 0.7, and 1, GPT-4 demonstrated a notable advantage over ChatGPT3.5, exhibiting a significantly higher total accuracy rate (Fig 1). Additional data analysis from the years 2021 and 2022 revealed GPT-4's consistent response to each question type across all temperature levels. This consistency indicates its reliability throughout the specified period and reflects its stable performance under varying temperature conditions (Fig 2).

Concerning variations in performance across question types, ChatGPT3.5 and GPT-4 exhibited distinct capabilities in answering specific questions. Specifically, ChatGPT3.5 specialise in responding to A1-type questions, while GPT-4 performs best on A3-type questions. Notably, both ChatGPT3.5 and GPT-4 demonstrated a similar weakness, scoring the lowest on X-type questions, providing insights into how the two models fare across diverse question types. This passage offers valuable insight into the divergences and similarities between the two models across various cognitive domains (Fig 3). The performance of ChatGPT remains relatively consistent across A1, A2, A3, and B-type questions, with only marginal effects observed from adjusting the temperature in relation to each question type. However, substituting ChatGPT3.5 with GPT-4 in all temperature settings substantially enhances accuracy across all question types (Fig 4).

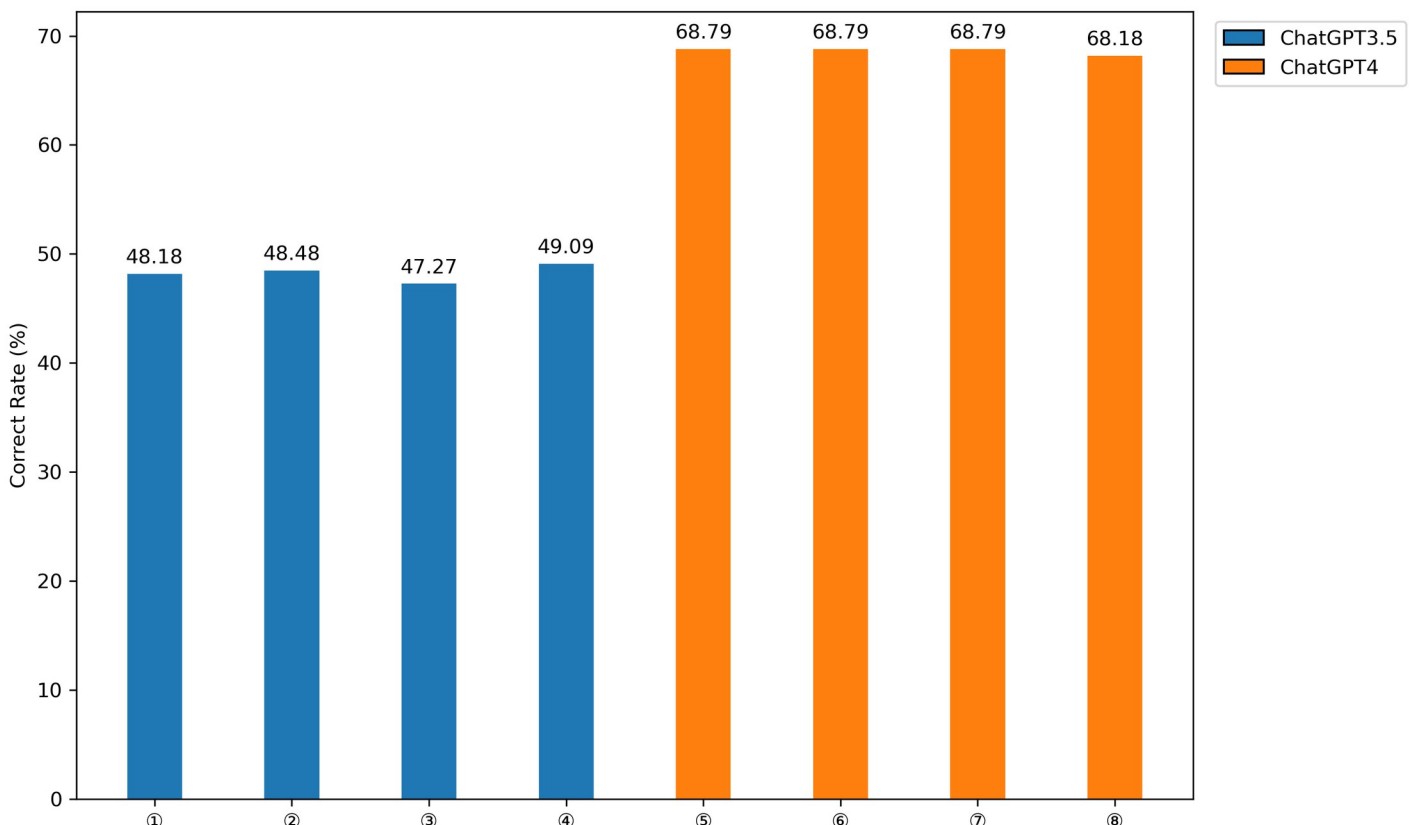

**Fig 1. The performance of ChatGPT at various temperatures.** (①:ChatGPT3.5(value 0),②:ChatGPT3.5(value 0.3),③:ChatGPT3.5(value 0.7), ④:ChatGPT3.5(value 1),⑤: GPT-4(value 0),⑥:GPT-4(value 0.3),⑦:GPT-4(value 0.7), ⑧:GPT-4(value 1)).

ChatGPT's exceptional performance in the medical humanities section underscores its proficiency in managing medical ethics and humanistic care. Simultaneously, its comparable performance in the remaining five sections demonstrates its ability to handle multidisciplinary knowledge domains in a balanced manner (Fig 5). It is noteworthy that both ChatGPT3.5 and GPT-4 exhibited identical accuracy rates in the medical humanities section at all four temperatures, whereas the correct rates fluctuated to some degree in the remaining five sections with changes in temperature (Fig 6).

In addition to the statistics on accuracy rates, we calculated the scores achieved by both ChatGPT3.5 and GPT-4 in the years 2021 and 2022, including the overall score, which was higher for GPT-4 than for ChatGPT3.5 (Fig 7). Furthermore, we analyzed of the violation of the answer rule in X-type multiple-choice questions. The higher percentage for ChatGPT3.5 (26.67%) compared to the lower percentage for GPT-4 (10%) underscores the latter's superiority in adhering to normative questions. The outcomes of this analysis, which delved into the specific behaviors behind the data, contribute to our understanding of the advantages of ChatGPT3.5 and GPT-4 in comprehensive medical examinations and provide insights into future directions for improvement.

## 4. Discussion

In China, the qualifying examination for admission to the Master's program in clinical medicine encompasses Comprehensive Clinical Medicine, Politics, and English—three subjects

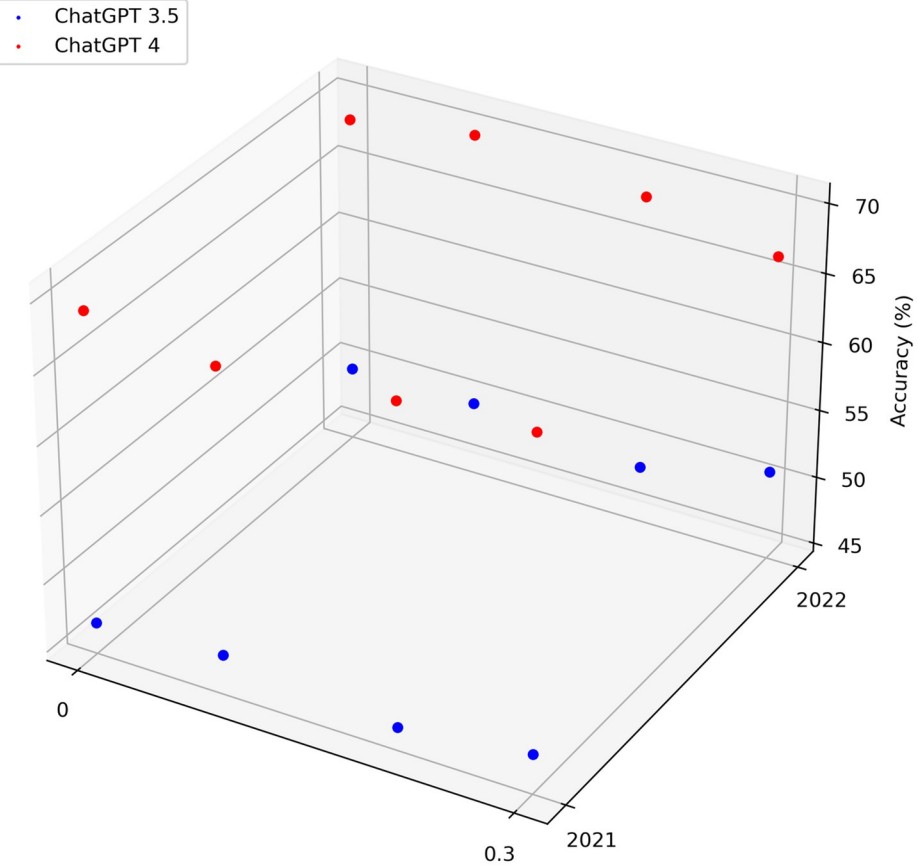

**Fig 2. Correctness of ChatGPT3.5 and GPT-4 in different years at different temperatures.**

under investigation in this research. The national authorities determine the minimum thresholds for each examination and the total score, considering the test question difficulty and the average scores of candidates in the current year. Candidates are required to meet the corresponding thresholds for each subject and the total score to fulfill the admission requirements. To ensure educational equity, the government categorizes universities into Zone A and Zone B based on the economic development and educational resources of each province. This categorization leads to distinct admission requirements, as the economic and educational levels of Zone A are generally more advanced. Consequently, the admission scores for Zone A are higher than those for Zone B.

Official data reveals that the score threshold for the 2021 Clinical Medicine General Examination was 123 in Zone A and 114 in Zone B. Revised data from 2022 reports an increased score threshold of 129 in Zone A and 120 in Zone B. During the 2021 test questions, ChatGPT3.5 scored the lowest (135 points) at a temperature value of 0.7, while GPT-4 scored the highest (208 points) at a temperature of one. On the 2022 test, ChatGPT3.5 achieved the lowest score of 144.5 points at a temperature value of 0.7, while GPT-4 achieved the highest score of 209 points at a temperature of 0.3. In comparison with the national minimum score, ChatGPT met the eligibility requirements for admission at any temperature. Bhayana and colleagues [12] reported positive results for ChatGPT in the Royal College of Canada Diagnostic Radiology Examination. Similarly, Fuentes-Martín and colleagues [13] found that ChatGPT

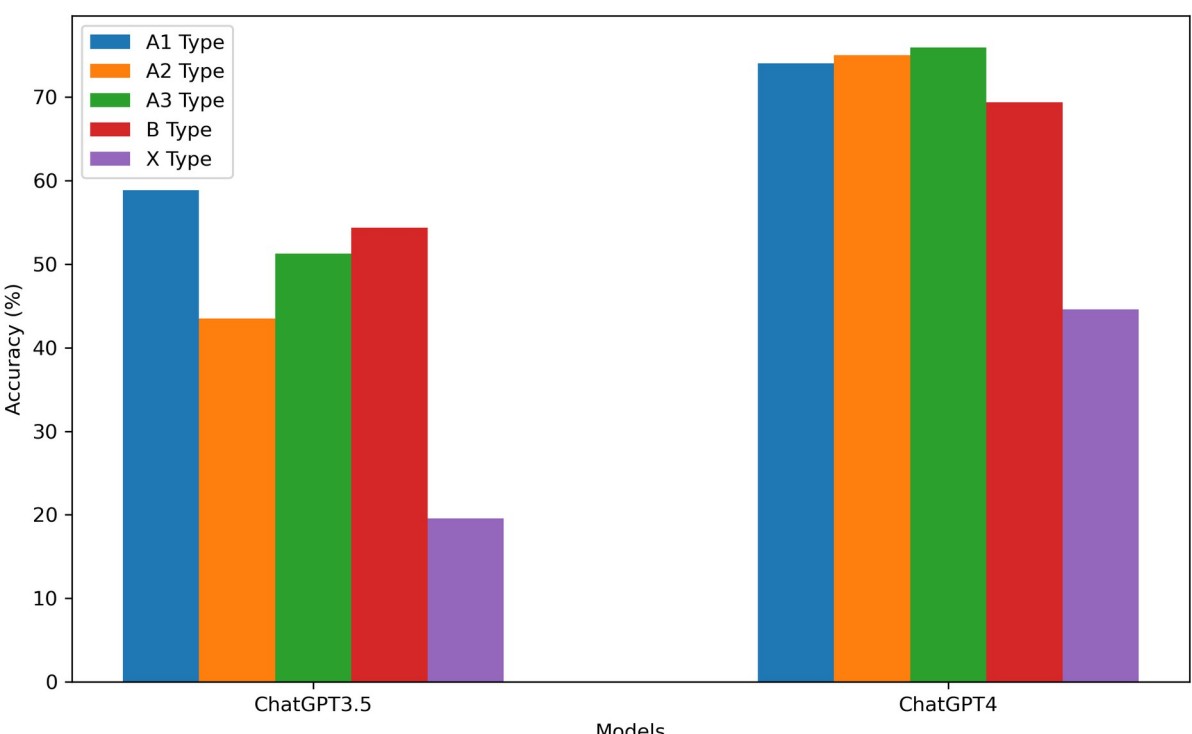

**Fig 3. Correctness of ChatGPT3.5 and GPT-4 on different question types.**

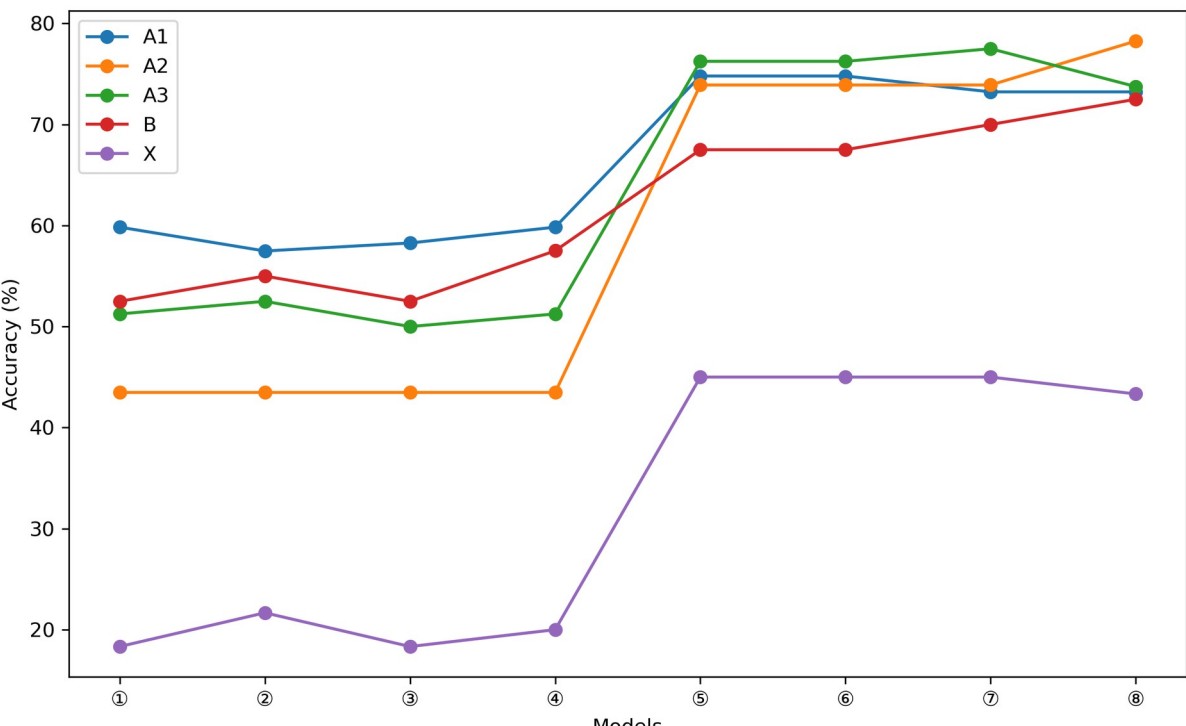

**Fig 4. Correctness of ChatGPT3.5 and GPT-4 on different question types at different temperatures.** (①:ChatGPT3.5(value 0),②: ChatGPT3.5(value 0.3),③:ChatGPT3.5(value 0.7), ④:ChatGPT3.5(value 1),⑤: GPT-4(value 0),⑥:GPT-4(value 0.3),⑦:GPT-4(value 0.7), ⑧: GPT-4(value 1)).

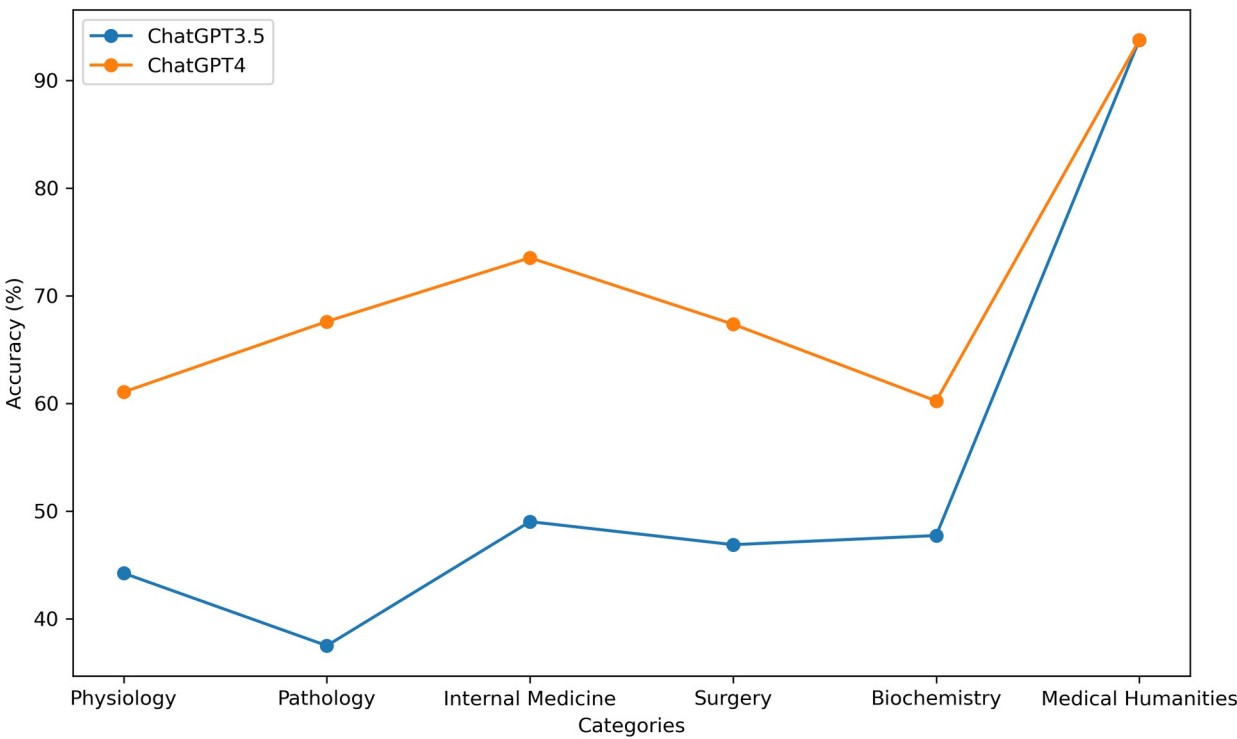

**Fig 5. Correctness of ChatGPT3.5 and GPT-4 on different subjects.**

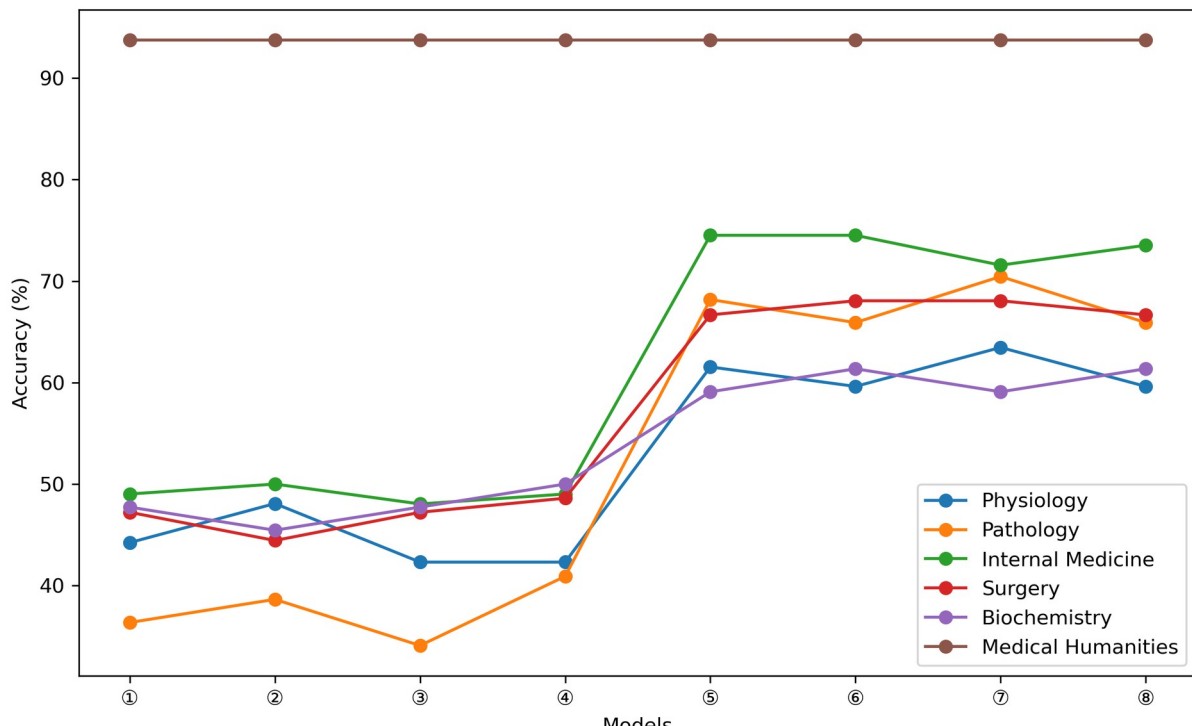

**Fig 6. Correctness of ChatGPT3.5 and GPT-4 on different subjects at different temperatures.** (①:ChatGPT3.5(value 0),②:ChatGPT3.5 (value 0.3),③:ChatGPT3.5(value 0.7), ④:ChatGPT3.5(value 1),⑤: GPT-4(value 0),⑥:GPT-4(value 0.3),⑦:GPT-4(value 0.7), ⑧:GPT-4(value 1)).

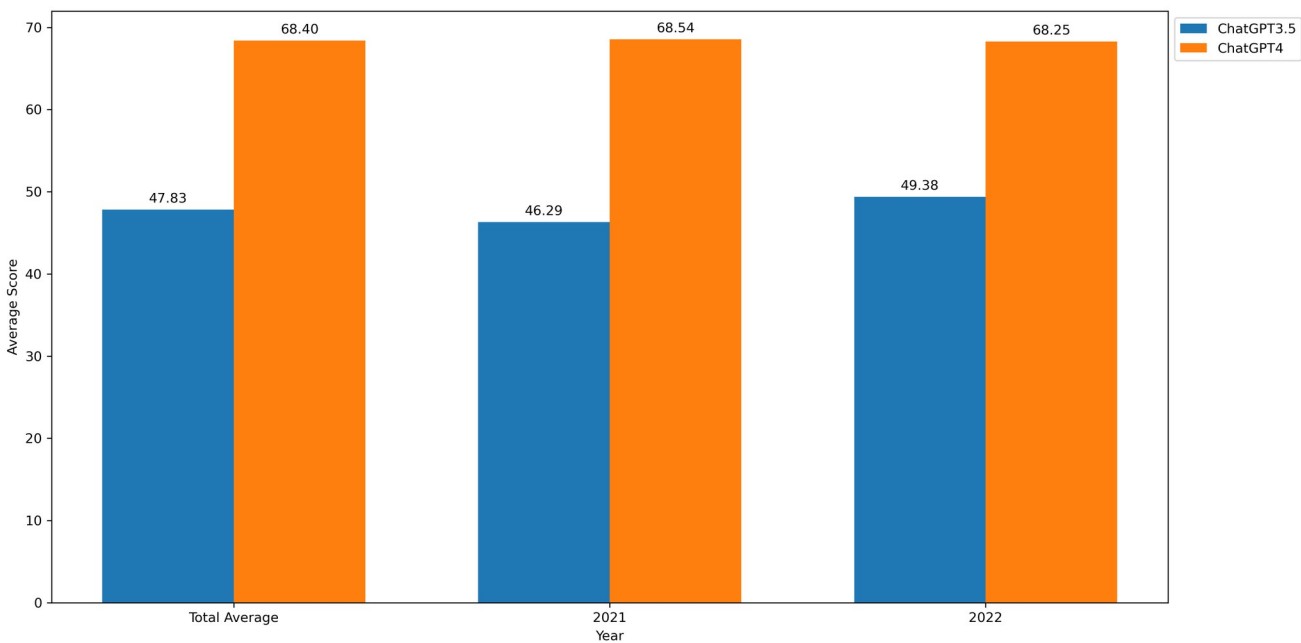

**Fig 7. Overall score rate, 2021 score rate, 2022 score rate for ChatGPT3.5 vs GPT-4.**

successfully passed the Andalusian Health Authority's 2022 Competitive Examination for Thoracic Surgery Specialist Positions. These findings provide robust evidence supporting the successful implementation of ChatGPT.

When comparing the accuracy rates of ChatGPT3.5 and GPT-4 across different years, a noticeable improvement of approximately 20% is evident in the latter following its upgrade from ChatGPT3.5. Moreover, the extreme deviation of GPT-4 (0.61%) is smaller than that of ChatGPT3.5 (2.88%). These enhancements in accuracy and stability for GPT-4 signify OpenAI's successful training and upgrading of the model, as highlighted in the source [14, 15].

In our comprehensive assessesment of ChatGPT, it becomes evident that variations exist in its responses to different types of queries. ChatGPT3.5 demonstrates relatively precise performance in answering explicit inquiries based on factual information, whereas GPT-4 exhibits superior performance in handling case-based questions, which demand a higher level of comprehensive judgment. However, when addressing intricate queries of type X, ChatGPT3.5 and GPT-4 display insufficient logical and critical thinking abilities, indicating the need for further refinement in forthcoming iterations and upgrades. This observation aligns with Miao 's research [8], suggesting that ChatGPT excels at handling straightforward fact-based inquiries but remains inadequate when confronted with demanding inquiries requiring profound understanding and meticulous computations. This study underscores the challenges encountered in utilizing ChatGPT in the medical field and sets the stage for future technological advancements to facilitate a more all-encompassing and comprehensive adaptation of ChatGPT to the varied task demands specific to the medical domain.

Both ChatGPT3.5 and GPT-4 demonstrated a high level of accuracy (93.75%) in the medical humanities field, securing the top position among various disciplines. This accuracy showcases their proficiency not only in professional knowledge but also in effective communication with patients, along with their familiarity with medical ethics and relevant regulations.

However, ChatGPT's performance varied across sub-disciplines. ChatGPT3.5 exhibited the lowest accuracy percentage in Pathology (37.5%), while GPT-4 was least proficient in Biochemistry (60.23%). Notably, GPT-4 displayed the most significant improvement in Pathology (30.11%) compared to ChatGPT3.5. The pathology section primarily comprised A1 and X-type questions, which were relatively uncomplicated and direct in nature, primarily evaluating candidates' ability to recall relevant information without necessitating complex logical reasoning and analysis. Consequently, once GPT-4 has been sufficiently trained with more data and has developed a more extensive foundation of background knowledge, it will be better equipped to assesses and answer these types of questions with greater precision. At this stage, when students utilize ChatGPT for revision and exam preparation, it is essential for them to engage in independent thought and critical evaluation of the provided answers. Blindly relying on ChatGPT is discouraged. Specifically, caution is advised when considering ChatGPT's responses and insights for X-type questions and those pertaining to the Pathology and Biochemistry sections. Throughout the utilization of ChatGPT, students must apply their own logical reasoning and judgment to verify the accuracy and reliability of the information and answers, thereby enhancing the learning outcomes and preparation quality. In future ChatGPT training sessions, students should address their skill gaps by intensifying their training for A2-type questions, X-type questions, as well as the Pathology and Biochemistry sections. This approach will contribute to enhancing the accuracy and reliability of ChatGPT in these specific areas, rendering it more comprehensive and effective in handling intricate medical knowledge and question types.

Compared to ChatGPT3.5, GPT-4 demonstrated a reduced probability of answering all questions incorrectly (26.36% < 48.18%) and an increased likelihood of answering all questions correctly (64.24% > 43.94%) across all four temperatures. In X-type questions, GPT-4 exhibited a lower rate of anomalous single-option responses (10.00% < 26.67%). This highlights that GPT-4's performance is significantly enhanced when adhering to user-prescribed responses. The resulting implications are highly promising for practical, high-accuracy applications, such as clinical medicine, presenting a more dependable and efficient option for intelligent medical assistants. Furthermore, students may utilise ChatGPT to obtain solutions and explanations for these medical examination questions, aiding in the consolidation of knowledge and enhancement of problem-solving skills. It is heartening to observe that ChatGPT has attained the minimum requirements for admission to the Chinese Master's Degree Entrance Qualification Examination in Clinical Medicine without undergoing specific training tailored to target the test questions.

The study not only demonstrates the current capabilities of ChatGPT but also highlights its strengths and weaknesses, offering valuable insights for the future training of large language models in the medical domain. Concurrently, the study results underscore the importance of maintaining critical thinking when utilizing ChatGPT. Users should remain cognizant of the fact that its responses may not always be accurate, emphasizing the need to avoid blind reliance and exercise caution. This is a crucial aspect that demands our ongoing attention.

This study has several limitations. Firstly, the test questions used were dated 27th December 2020 and 26th December 2021, while the databases within ChatGPT3.5 and GPT-4 were last updated in January 2022. This creates uncertainty regarding whether OpenAI used these specific test questions for model training during ChatGPT's training. Therefore, there exists a risk that the accuracy rate of ChatGPT, as determined by this test, may be artificially inflated. Secondly, although the study discovered that adjusting the temperature did not significantly impact the accuracy rate, a comprehensive summary of the temperature's impact on the accuracy rate remains challenging. More experimental data is needed for further scrutiny. Thirdly, due to the incomplete disclosure of admission scores by each medical school, it

was impossible to compare the ChatGPT scores with the admission score lines to determine whether they met specific school admission requirements. Fourth, as an advanced artificial intelligence tool, ChatGPT's mechanisms of logical operation and decision-making in the process of analyzing matters and making decisions cannot always be fully understood by humans. Therefore, when applying ChatGPT to clinical practice, it is crucial to ensure that experienced doctors are involved and supervise the entire process [16]. Furthermore, the test questions used in this study were written in Chinese, while ChatGPT was primarily trained in English. Nuanced discrepancies in grammar rules and other aspects between the Chinese and English languages might affect ChatGPT's effectiveness when used with Chinese. The current performance is restricted by the corpus, and further optimization and adjustment are required [17]. Consequently, the findings of this study provide an incomplete representation of ChatGPT's overall performance level. However, it is anticipated that with more training on a Chinese corpus, the performance of ChatGPT will be further enhanced. Despite these limitations, our study is the first study assessesed the reliability and utility of ChatGPT in the field of medical education in China. Furthermore, our study provides insightful knowledge on using AI in medical field.

## 5. Conclusion

This study innovatively tested the performance of ChatGPT in the Chinese Master's Degree Entrance Examination in Clinical Medicine, filling a knowledge gap in the intersection of medical education and artificial intelligence. While ChatGPT has met the requirements for passing the Chinese Master's Comprehensive Examination in Clinical Medicine, it still fails to respond accurately to approximately 37% of the questions, indicating potential hazards of incorrect judgments in a healthcare environment. Medical students and clinicians should exercise caution when using ChatGPT, recognizing its imperfections and limitations. Simultaneously, when utilised, providing more detailed contextual information and relevant knowledge can enhance the accuracy of its responses [18]. Therefore, ChatGPT must continuously develop and enhance its precision to meet the increasingly rigorous clinical requirements. The strict supervision of medical professionals is pivotal at the end of the AI processing chain, ensuring ChatGPT's safe and reliable application.

Furthermore, to better transform clinical practice and medical education, a closer collaboration between artificial intelligence companies and clinical practitioners is essential. This entails the meticulous development of specialised training corpora and the concerted effort to develop a medical professional version of ChatGPT [19]. Such initiatives could address the limitations identified in this study, thereby enhancing the effectiveness of support in medical education and healthcare sectors with greater precision and relevance.

## Supporting information

**S1 Data.**
(XLS)

## Acknowledgments

All authors have made significant contributions to this study and the field of medical education. As we reach the completion of this paper, I would like to express my sincere gratitude and extend my best wishes to those who have supported and guided me throughout this research and learning journey.

## Author Contributions

**Data curation:** Md. Shahjalal, Zi-Fan Zhuang.

**Software:** Bai-Xiang He, Chen Li.

**Writing – original draft:** Ke-Cheng Li, Zhi-Jun Bu, Zhao-Lan Liu.

**Writing – review & editing:** Jian-Ping Liu, Bin Wang, Zhao-Lan Liu.

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
