## [Decision Letter · Decision Letter 0]

11 Mar 2024

PONE-D-24-06964Performance of ChatGPT on Chinese Master's Degree Entrance Examination in Clinical MedicinePLOS ONE

Dear Dr. Liu,

Thank you for submitting your manuscript to PLOS ONE. After careful consideration, we feel that it has merit but does not fully meet PLOS ONE’s publication criteria as it currently stands. Therefore, we invite you to submit a revised version of the manuscript that addresses the points raised during the review process.

**ACADEMIC EDITOR: ** **Following are recommendations:**1. ChatGPT 4 cannot process image input and assist with image interpretation directly. Were there any questions with images? If yes, how many?

2. Please look at this article to discuss limitations and ethical concerns as that would add to the study. (https://www.cureus.com/articles/161200-radiology-gets-chatty-the-chatgpt-saga-unfolds#!/). AI ethics is important as AI decision are not always intelligible to humans.

3. There have been a few articles about use of ChatGPT in medical examinations. In one of them, they compared ChatGPT to instructGPT3 and chatgpt did outperform it (https://www.ncbi.nlm.nih.gov/pmc/articles/PMC9947764/). Do you have any such comparisons?

4. The abstract could be more succinct. In the Results section, the comparison between ChatGPT3.5 and GPT-4 could be presented in a more structured manner. Consider organizing the information chronologically or by thematic relevance to facilitate a smoother flow for the reader.

5. Furthermore, the Conclusion could be strengthened by offering specific recommendations for the identified enhancements in ChatGPT. Line 345. ‘Furthermore, the potential of close collaboration between AI companies and clinicians must not be overemphasized, ‘ Can you please elaborate on that ? **Following are suggestions but not hard recommendations however I feel they will certainly enhance the clarity and strength of the study:**

1.The introduction could provide more background information on the Chinese Master's Degree Entrance Examination in Clinical Medicine to help readers understand the context and significance of the study.

2.The methods section should provide more details on how the researchers inputted the questions into ChatGPT and how they recorded and verified the responses.

3.Discussion should include what are some concrete ways ChatGPT could be leveraged to assist students in exam preparation?

4.The limitations of the study could be expanded. For example, the authors note the discrepancy in time between when the test questions were administered vs. when ChatGPT's databases were last updated, but they could discuss more how this may have impacted the results.

5. Conclusion section should address some areas where knowledge gap was filled or knowledge was advanced with this study. Please ensure that your decision is justified on PLOS ONE’s publication criteria and not, for example, on novelty or perceived impact.

We look forward to receiving your revised manuscript.

Kind regards,

Harpreet Singh Grewal

Academic Editor

PLOS ONE

Journal Requirements:

“This study is supported by a grant from the National Natural Science Foundation of China (Grant No. 82374298 ) and the Reserve Discipline Leader Funding of Beijing University of Chinese Medicine (Grant No. 90010960920033).”

4. Please be informed that funding information should not appear in the Acknowledgments section or other areas of your manuscript. We will only publish funding information present in the Funding Statement section of the online submission form. Please remove any funding-related text from the manuscript.

5. In the online submission form you indicate that your data is not available for proprietary reasons and have provided a contact point for accessing this data. Please note that your current contact point is a co-author on this manuscript. According to our Data Policy, the contact point must not be an author on the manuscript and must be an institutional contact, ideally not an individual. Please revise your data statement to a non-author institutional point of contact, such as a data access or ethics committee, and send this to us via return email. Please also include contact information for the third party organization, and please include the full citation of where the data can be found.

6. Please include a separate caption for each figure in your manuscript.

Additional Editor Comments (if provided):

Here are a few points/questions

1. ChatGPT 4 cannot process image input and assist with image interpretation directly. Were there any questions with images? If yes, how many?

2. Please look at this article to discuss limitations and ethical concerns as that would add to the study. (https://www.cureus.com/articles/161200-radiology-gets-chatty-the-chatgpt-saga-unfolds#!/). AI ethics is important as AI decision are not always intelligible to humans.

3. There have been a few articles about use of ChatGPT in medical examinations. In one of them, they compared ChatGPT to instructGPT3 and chatgpt did outperform it (https://www.ncbi.nlm.nih.gov/pmc/articles/PMC9947764/). Do you have any such comparisons?

4. The abstract could be more succinct. In the Results section, the comparison between ChatGPT3.5 and GPT-4 could be presented in a more structured manner. Consider organizing the information chronologically or by thematic relevance to facilitate a smoother flow for the reader.

5. Furthermore, the Conclusion could be strengthened by offering specific recommendations for the identified enhancements in ChatGPT. Line 345. ‘Furthermore, the potential of close collaboration between AI companies and clinicians must not be overemphasized, ‘ Can you please elaborate on that

6.The introduction could provide more background information on the Chinese Master's Degree Entrance Examination in Clinical Medicine to help readers understand the context and significance of the study.

7.The methods section should provide more details on how the researchers inputted the questions into ChatGPT and how they recorded and verified the responses.

8.Discussion should include what are some concrete ways ChatGPT could be leveraged to assist students in exam preparation?

9.The limitations of the study could be expanded. For example, the authors note the discrepancy in time between when the test questions were administered vs. when ChatGPT's databases were last updated, but they could discuss more how this may have impacted the results.

10. Conclusion section should address some areas where knowledge gap was filled or knowledge was advanced with this study.

Reviewers' comments:

Reviewer's Responses to Questions

**Comments to the Author**

1. Is the manuscript technically sound, and do the data support the conclusions?

Reviewer #1: Yes

Reviewer #2: Yes

2. Has the statistical analysis been performed appropriately and rigorously? 

Reviewer #1: Yes

Reviewer #2: N/A

3. Have the authors made all data underlying the findings in their manuscript fully available?

Reviewer #1: Yes

Reviewer #2: Yes

4. Is the manuscript presented in an intelligible fashion and written in standard English?

Reviewer #1: Yes

Reviewer #2: Yes

5. Review Comments to the Author

Reviewer #1: The article provides a comprehensive evaluation of ChatGPT's reliability and utility in the realm of medical education, using the Chinese Clinical Medicine Master's Entrance Examination as a performance benchmark. Great job with the article. While the study sheds light on the strengths and weaknesses of ChatGPT, there are areas that could benefit from refinement to enhance the manuscript's clarity and precision.

Here are a few points/questions

1. ChatGPT 4 cannot process image input and assist with image interpretation directly. Were there any questions with images? If yes, how many?

2. Please look at this article to discuss limitations and ethical concerns as that would add to the study. (https://www.cureus.com/articles/161200-radiology-gets-chatty-the-chatgpt-saga-unfolds#!/). AI ethics is important as AI decision are not always intelligible to humans.

3. There have been a few articles about use of ChatGPT in medical examinations. In one of them, they compared ChatGPT to instructGPT3 and chatgpt did outperform it (https://www.ncbi.nlm.nih.gov/pmc/articles/PMC9947764/). Do you have any such comparisons?

4. The abstract could be more succinct. In the Results section, the comparison between ChatGPT3.5 and GPT-4 could be presented in a more structured manner. Consider organizing the information chronologically or by thematic relevance to facilitate a smoother flow for the reader.

5. Furthermore, the Conclusion could be strengthened by offering specific recommendations for the identified enhancements in ChatGPT. Line 345. ‘Furthermore, the potential of close collaboration between AI companies and clinicians must not be overemphasized, ‘ Can you please elaborate on that

In summary, while the study presents valuable insights, refining the abstract for brevity, enhancing the organization of results, providing additional context for variations, and offering specific improvement recommendations in the conclusion will contribute to a more polished and impactful manuscript.

Reviewer #2: The introduction could provide more background information on the Chinese Master's Degree Entrance Examination in Clinical Medicine to help readers understand the context and significance of the study.

The methods section should provide more details on how the researchers inputted the questions into ChatGPT and how they recorded and verified the responses.

The discussion could delve deeper into the implications of the findings for medical education and the potential role of AI language models like ChatGPT in this field. What are some concrete ways ChatGPT could be leveraged to assist students in exam preparation?

The limitations of the study could be expanded. For example, the authors note the discrepancy in time between when the test questions were administered vs. when ChatGPT's databases were last updated, but they could discuss more how this may have impacted the results.

The conclusion would benefit from more specific recommendations for future research directions to address the identified limitations and knowledge gaps.

Cohesion:

the first sentence of the 4th paragraph in the Discussion seems abrupt.

The organization of the Discussion section could be improved by using subheadings to clearly delineate the different topics covered (e.g. comparison of ChatGPT versions, performance across question types, implications, limitations).

Grammar:

"This study assess" should be "This study assesses"

"their efficacy in meeting user needs is managing" should perhaps be "their efficacy in meeting user needs in managing"

6. PLOS authors have the option to publish the peer review history of their article (what does this mean?). If published, this will include your full peer review and any attached files.

Reviewer #1: **Yes: **Gagandeep Dhillon

Reviewer #2: **Yes: **Ankit Virmani

---

## [Author Response · Author response to Decision Letter 0]

18 Mar 2024

Dear Editors and Reviewers of the PLOS ONE, 

On behalf of all the co-authors, I would like to express our profound gratitude for the 

diligent work you have invested in the review process. We fully acknowledge the 

difficulty of this task and extend our highest regards for your professionalism and 

dedication. 

After receiving your valuable feedback, our team engaged in thorough discussions 

and made extensive revisions to our manuscript accordingly. 

Furthermore, we would like to take this opportunity to convey our heartfelt new year wishes to you all, following the recent Lunar New Year celebrations. May the coming 

year bring you prosperity, success, and health. At the end of this letter, we have attached a detailed response to each review comment and the corresponding revisions 

made. Once again, we express our deepest appreciation for your invaluable contributions. 

Yours sincerely, 

Zhao-Lan Liu 

Beijing University of Chinese Medicine

Respond

ACADEMIC EDITOR:

Following are recommendations:

1. ChatGPT 4 cannot process image input and assist with image interpretation directly. Were there any questions with images? If yes, how many?

2. Please look at this article to discuss limitations and ethical concerns as that would add to the study. (https://www.cureus.com/articles/161200-radiology-gets-chatty-the-chatgpt-saga-unfolds#!/). AI ethics is important as AI decision are not always intelligible to humans.

3. There have been a few articles about use of ChatGPT in medical examinations. In one of them, they compared ChatGPT to instructGPT3 and chatgpt did outperform it (https://www.ncbi.nlm.nih.gov/pmc/articles/PMC9947764/). Do you have any such comparisons?

4. The abstract could be more succinct. In the Results section, the comparison between ChatGPT3.5 and GPT-4 could be presented in a more structured manner. Consider organizing the information chronologically or by thematic relevance to facilitate a smoother flow for the reader.

5. Furthermore, the Conclusion could be strengthened by offering specific recommendations for the identified enhancements in ChatGPT. Line 345. ‘Furthermore, the potential of close collaboration between AI companies and clinicians must not be overemphasized, ‘ Can you please elaborate on that ?

Following are suggestions but not hard recommendations however I feel they will certainly enhance the clarity and strength of the study:

1.The introduction could provide more background information on the Chinese Master's Degree Entrance Examination in Clinical Medicine to help readers understand the context and significance of the study.

2.The methods section should provide more details on how the researchers inputted the questions into ChatGPT and how they recorded and verified the responses.

3.Discussion should include what are some concrete ways ChatGPT could be leveraged to assist students in exam preparation?

4.The limitations of the study could be expanded. For example, the authors note the discrepancy in time between when the test questions were administered vs. when ChatGPT's databases were last updated, but they could discuss more how this may have impacted the results.

5. Conclusion section should address some areas where knowledge gap was filled or knowledge was advanced with this study.

Q1.ChatGPT 4 cannot process image input and assist with image interpretation directly. Were there any questions with images? If yes, how many?

Respond to Q1

Dear Reviewer, First and foremost, we would like to express our sincere gratitude for your meticulous review and your professional inquiry.Indeed, as you correctly pointed out, GPT-4 does not possess the capability to directly process or interpret image inputs. In alignment with this limitation, I can confirm that none of the questions we utilised involved any images, nor did they assess the examinees' ability to recognise or interpret images. Hence, concerns pertaining to image handling are not applicable in our context. We appreciate your vigilance in ensuring the integrity and relevance of our methodology.

Q2. Please look at this article to discuss limitations and ethical concerns as that would add to the study. (https://www.cureus.com/articles/161200-radiology-gets-chatty-the-chatgpt-saga-unfolds#!/). AI ethics is important as AI decision are not always intelligible to humans.

Respond to Q2

Dear Reviewer, thank you for your invaluable suggestion to delve into the limitations and ethical concerns surrounding artificial intelligence, especially considering the intelligibility of AI decisions to humans. Following your advice, we have thoroughly reviewed and cited the article you mentioned. Consequently, we have enriched our discussion section with an in-depth analysis of ChatGPT's limitations and the ethical considerations of AI. This addition not only enhances the content of our paper but also deepens the discourse on these crucial topics.We appreciate your guidance in making our study more comprehensive and thought-provoking.

Modify Details：

Fourth, as an advanced artificial intelligence tool, ChatGPT's mechanisms of logical operation and decision-making in the process of analyzing matters and making decisions cannot always be fully understood by humans. Therefore, when applying ChatGPT to clinical practice, it is crucial to ensure that experienced doctors are involved and supervise the entire process[16].

[16]Grewal H, Dhillon G, Monga V, et al. Radiology Gets Chatty: The ChatGPT Saga Unfolds. Cureus. 2023 Jun 8;15(6):e40135. 

Page:16 Line:349-354

Q3. There have been a few articles about use of ChatGPT in medical examinations. In one of them, they compared ChatGPT to instructGPT3 and chatgpt did outperform it (https://www.ncbi.nlm.nih.gov/pmc/articles/PMC9947764/). Do you have any such comparisons?

Respond to Q3

Dear Reviewer, thank you for your insightful comments and for directing our attention to the recent articles comparing the use of ChatGPT in medical examinations, including its comparison with InstructGPT. Following your advice, we diligently reviewed the article you recommended and further explored the relevant literature to enhance our understanding.

InstructGPT is trained using a technique known as "contrastive learning," which involves comparing different responses generated by the model and selecting the one that best fits a given instruction. This process inherently involves human judgement in choosing the most appropriate answer provided by ChatGPT, a factor we aimed to minimize in our study. Our research was designed to focus more on the accuracy of large language models like ChatGPT in handling medical questions without professional training or human intervention.

Therefore, our study primarily concentrates on the comparison between ChatGPT and GPT-4, rather than including InstructGPT. This approach was chosen to better understand the capabilities and limitations of these models in a more controlled and less human-influenced context.

We hope this clarification addresses your query and appreciate your guidance in refining our study.

Q4. The abstract could be more succinct. In the Results section, the comparison between ChatGPT3.5 and GPT-4 could be presented in a more structured manner. Consider organizing the information chronologically or by thematic relevance to facilitate a smoother flow for the reader.

Respond to Q4

Dear Reviewer, thank you for your thorough review and constructive feedback. We acknowledge that the abstract was indeed somewhat verbose and have since revised it to be more concise. 

The results section has been reorganized to present our findings in a structured manner, categorizing them based on several thematic aspects: the impact of temperature settings on ChatGPT's performance, the influence of different types of multiple-choice questions on ChatGPT, the effect of various knowledge domains on ChatGPT's accuracy, and ChatGPT's performance across different years. This reorganization aims to provide a clearer and more coherent flow of information, facilitating an easier comprehension for the reader.

Furthermore, we have added distinct titles to each figure in the article, enhancing the clarity and readability of our findings.

We greatly appreciate your guidance in improving the quality and readability of our work.

Modify Details：

Background: ChatGPT is a large language model designed to generate responses based on a contextual understanding of user queries and requests.This study utilised the entrance examination for the Master of Clinical Medicine in Traditional Chinese Medicine to assess the reliability and practicality of ChatGPT within the domain of medical education.

Methods: We selected 330 single and multiple-choice questions from the 2021 and 2022 Chinese Master of Clinical Medicine comprehensive examinations, which did not include any images or tables. To ensure the test's accuracy and authenticity, we preserved the original format of the query and alternative test texts, without any modifications or explanations.

Page:2 Line:29-37

At temperatures of 0, 0.3, 0.7, and 1, GPT-4 demonstrated a notable advantage over ChatGPT3.5, exhibiting a significantly higher total accuracy rate (Figure 1). Additional data analysis from the years 2021 and 2022 revealed GPT-4's consistent response to each question type across all temperature levels. This consistency indicates its reliability throughout the specified period and reflects its stable performance under varying temperature conditions (Figure 2). 

Figure 1. The Performance of ChatGPT at Various Temperatures.

(①:ChatGPT3.5(value 0),②:ChatGPT3.5(value 0.3),③:ChatGPT3.5(value 0.7), ④:ChatGPT3.5(value 1),⑤: GPT-4(value 0),⑥:GPT-4(value 0.3),⑦:GPT-4(value 0.7), ⑧:GPT-4(value 1))

Figure 2. Correctness of ChatGPT3.5 and GPT-4 in different years at different temperatures.

Concerning variations in performance across question types, ChatGPT3.5 and GPT-4 exhibited distinct capabilities in answering specific questions. Specifically, ChatGPT3.5 specialise in responding to A1-type questions, while GPT-4 performs best on A3-type questions. Notably, both ChatGPT3.5 and GPT-4 demonstrated a similar weakness, scoring the lowest on X-type questions, providing insights into how the two models fare across diverse question types. This passage offers valuable insight into the divergences and similarities between the two models across various cognitive domains (Figure 3). The performance of ChatGPT remains relatively consistent across A1, A2, A3, and B-type questions, with only marginal effects observed from adjusting the temperature in relation to each question type. However, substituting ChatGPT3.5 with GPT-4 in all temperature settings substantially enhances accuracy across all question types (Figure 4).

Figure 3. Correctness of ChatGPT3.5 and GPT-4 on different question types.

Figure 4.Correctness of ChatGPT3.5 and GPT-4 on different question types at different temperatures.

(①:ChatGPT3.5(value 0),②:ChatGPT3.5(value 0.3),③:ChatGPT3.5(value 0.7), ④:ChatGPT3.5(value 1),⑤: GPT-4(value 0),⑥:GPT-4(value 0.3),⑦:GPT-4(value 0.7), ⑧:GPT-4(value 1))

ChatGPT's exceptional performance in the medical humanities section underscores its proficiency in managing medical ethics and humanistic care. Simultaneously, its comparable performance in the remaining five sections demonstrates its ability to handle multidisciplinary knowledge domains in a balanced manner (Figure 5). It is noteworthy that both ChatGPT3.5 and GPT-4 exhibited identical accuracy rates in the medical humanities section at all four temperatures, whereas the correct rates fluctuated to some degree in the remaining five sections with changes in temperature (Figure 6).

Figure 5.Correctness of ChatGPT3.5 and GPT-4 on different subjects.

Figure 6.Correctness of ChatGPT3.5 and GPT-4 on different subjects at different temperatures.

(①:ChatGPT3.5(value 0),②:ChatGPT3.5(value 0.3),③:ChatGPT3.5(value 0.7), ④:ChatGPT3.5(value 1),⑤: GPT-4(value 0),⑥:GPT-4(value 0.3),⑦:GPT-4(value 0.7), ⑧:GPT-4(value 1))

In addition to the statistics on accuracy rates, we calculated the scores achieved by both ChatGPT3.5 and GPT-4 in the years 2021 and 2022, including the overall score, which was higher for GPT-4 than for ChatGPT3.5 (Figure 7). Furthermore, we analyzed of the violation of the answer rule in X-type multiple-choice questions. The higher percentage for ChatGPT3.5 (26.67%) compared to the lower percentage for GPT-4 (10%) underscores the latter's superiority in adhering to normative questions. The outcomes of this analysis, which delved into the specific behaviors behind the data, contribute to our understanding of the advantages of ChatGPT3.5 and GPT-4 in comprehensive medical examinations and provide insights into future directions for improvement.

Figure 7.Overall Score Rate, 2021 Score Rate, 2022 Score Rate for ChatGPT3.5 vs GPT-4.

Page:9-11 Line:186-241

Q5. Furthermore, the Conclusion could be strengthened by offering specific recommendations for the identified enhancements in ChatGPT. Line 345. ‘Furthermore, the potential of close collaboration between AI companies and clinicians must not be overemphasized, ‘ Can you please elaborate on that ?

Respond to Q5

Dear Reviewer, thank you for your valuable feedback regarding our conclusion and the specific passage you referenced. We have taken your comments into consideration and have accordingly revised the conclusion to articulate specific improvements and recommendations for enhancing the accuracy of ChatGPT.

Regarding the line you mentioned, our apologies for any confusion caused by its previous vagueness. We have since rewritten this section for clarity. 

We hope this clarification and the revisions made throughout the manuscript better address your concerns and contribute to a clearer understanding of our study's implications and recommendations.

Modify Details：

Simultaneously, when utilised, providing more detailed contextual information and relevant knowledge can enhance the accuracy of its responses[18].

[18]Goodman RS, Patrinely JR, Stone CA Jr, et al. Accuracy and Reliability of Chatbot Responses to Physician Questions. JAMA Netw Open. 2023 Oct 2;6(10):e2336483.

Page:17 Line:373-375

Furthermore, to better transform clinical practice and medical education, a closer collaboration between artificial intelligence companies and clinical practitioners is essential. This entails the meticulous development of specialised training corpora and the concerted effort to develop a medical professional version of ChatGPT[19]. Such initiatives could address the limitations identified in this study, thereby enhancing the effectiveness of support in medical education and healthcare sectors with greater precision and relevance.

[19]Li J, Dada A, Puladi B, Kleesiek J, et al. ChatGPT in healthcare: A taxonomy and systematic review. Comput Methods Programs Biomed. 2024 Mar;245:108013. 

Page:18 Line:379-385

Q6.The introduction could provide more background information on the Chinese Master's Degree Entrance Examination in Clinical Medicine to help readers understand the context and significance of the study.

Respond to Q6

Dear Reviewer, thank you for your constructive feedback on the introduction of our manuscript. Acknowledging the importance of context for our international readers, we have expanded the introduction to include a detailed description of the examination's background, its significance in the field of medical education in China, and its role in assessing the competence and readiness of candidates aspiring to pursue a master's degree in clinical medicine.

We believe that these additions will greatly enhance the reader's comprehension of the study's context and the significance of our findings within the broader landscape of medical education and artificial intelligence.

Modify Details：

NMPUA represents a critically important examination for aspiring master's degree students, aimed at assessing the clinical reasoning, knowledge, diagnostic capabilities, and decision-making proficiency of medical undergraduates in a clinical context.

Page:5 Line:93-96

Q7.The methods section should provide more details on how the researchers inputted the questions into ChatGPT and how they recorded and verified the responses.

Respond to Q7

Dear Reviewer, thank you for your insightful feedback regarding the methods section of our manuscript. We recognize the importance of transparency and detail in describing our research process. Following your suggestion, we have now provided a comprehensive description of the procedures involved in inputting the questions into ChatGPT, as well as the methods we employed to record and verify the responses.

These enhancements to the methods section aim to provide readers with a clear and thorough understanding of our methodology, reinforcing the credibility and reliability of our findings.

We hope that these revisions adequately address your concerns and contribute to a more comprehensive and transparent presentation of our research process.

Modify Details：

In this study, we replicated and sent 330 multiple-choice questions from the 2021 and 2022 Chinese Master of Clinical Medicine comprehensive examinations to both ChatGPT3.5 and GPT-4, in the order they appeared in the examination papers. The request was for them to simulate the role of a doctor and provide answers accordingly.

Page:7 Line:137-140

Responses were meticulously recorded using Excel and cross-verified against the correct answers to ensure precise evaluation of their performance in the Masters Comprehensive Clinical Medicine Examination.

Page:7 Line:145-148

Q8.Discussion should include what are some concrete ways ChatGPT could be leveraged to assist students in exam preparation?

Respond to Q8

Dear Reviewer, thank you for your valuable suggestion to enhance the discussion section of our manuscript by including specific ways in which ChatGPT could be leveraged to assist students in their exam preparation. By including these concrete methods, we aim to provide readers with a clearer understanding of how AI, specifically ChatGPT, can be innovatively used to enrich medical education and examination preparation.

We appreciate your guidance, which has undoubtedly contributed to making our discussion more comprehensive and relevant to educators, students, and the broader academic community.

Modify Details：

Furthermore, students may utilise ChatGPT to obtain solutions and explanations for these medical examination questions, aiding in the consolidation of knowledge and enhancement of problem-solving skills.

Page:15 Line:325-327

Q9.The limitations of the study could be expanded. For example, the authors note the discrepancy in time between when the test questions were administered vs. when ChatGPT's databases were last updated, but they could discuss more how this may have impacted the results.

Respond to Q9

Dear Reviewer, thank you for your insightful recommendation to expand upon the limitations section of our study. We have taken your advice to heart and have elaborated on the discussion concerning the temporal discrepancies between the administration of the test questions and the last update of ChatGPT's databases.

By expanding the discussion on this limitation, we aim to provide a more comprehensive understanding of the factors that may influence the results of studies involving AI technologies in educational settings.

We appreciate your guidance in making our study more thorough and reflective of the complexities involved in integrating AI into educational contexts.

Modify Details：

Therefore, there exists a risk that the accuracy rate of ChatGPT, as determined by this test, may be artificially inflated.

Page:16 Line:342-343

Q10. Conclusion section should address some areas where knowledge gap was filled or knowledge was advanced with this study.

Respond to Q9

Dear Reviewer, thank you for your valuable feedback regarding the conclusion section of our manuscript. In response to your suggestion, we have carefully revised this section to underscore the significance of our study and the ways in which it has contributed to filling knowledge gaps and advancing knowledge within the relevant field.

These additions aim to provide a clear understanding of the contributions our study makes to the field, emphasizing its role in advancing our understanding of AI's potential and limitations in supporting medical education.

Modify Details：

This study innovatively tested the performance of ChatGPT in the Chinese Master's Degree Entrance Examination in Clinical Medicine, filling a knowledge gap in the intersection of medical education and artificial intelligence.

Page:17 Line:366-368

---

## [Editor Report · Decision Letter 1]

20 Mar 2024

ChatGPT在中国临床医学硕士入学考试中的表现

PONE-D-24-06964R1

Dear Dr. Liu,

We’re pleased to inform you that your manuscript has been judged scientifically suitable for publication and will be formally accepted for publication once it meets all outstanding technical requirements.

Kind regards,

Harpreet Singh Grewal

Academic Editor

PLOS ONE
---

## [Editor Report · Acceptance letter]

25 Mar 2024

PONE-D-24-06964R1 

PLOS ONE

Dear Dr. Liu, 

I'm pleased to inform you that your manuscript has been deemed suitable for publication in PLOS ONE. Congratulations! Your manuscript is now being handed over to our production team.

Kind regards, 

on behalf of

Dr. Harpreet Singh Grewal 

Academic Editor

PLOS ONE